# Symptoms of depression among outpatients with suspected COVID-19 in metropolitan Local Government Areas of Kaduna State, Nigeria

Gregory C. Umeh[1]*, Laurent Cleenwerck de Kiev[2], Jabani Mamza[1], Aliyu Atiku[1], Suleiman Mohammed[1], Dauda S. Hananiya[1], Moses Onoh[1], Habibu B. Yahaya[1], Basirat Adeoti[1], Rabiat T. Musa[1], Mutiu Adegbite[1], Sunday Audu[1], Jeremiah Daikwo[3], Neyu Iliyasu[4], Amina Mohammed Baloni[5]

**1** World Health Organization, Kaduna, Nigeria, **2** International Faculty Coordinator, Euclid University, Banjul, The Gambia, **3** Epidemiologist, Kaduna State Ministry of Health, Kaduna, Nigeria, **4** State Emergency Operations Centre (sEOC), Kaduna, Nigeria, **5** Kaduna State Ministry of Health, Kaduna, Nigeria

* umehg@who.int

**Data Availability Statement:** All relevant data are included via the dataset unique digital object identifier (DOI):doi:10.5061/dryad.wdbrv15wj.

## Abstract

### Background

The novel SARS-CoV-2 virus that causes Coronavirus disease (COVID-19) has redefined global health and response to Acute Respiratory Infection (ARI). The outbreak of a cluster of influenza-like illnesses in Wuhan, China, has morphed into a pandemic in the last quarter of 2019, stretching from South East Asia to Europe, The Americas, Africa, and the Australian subcontinent. We evaluated the prevalence of depression among outpatients diagnosed with ARI.

### Materials and methods

We utilized a cross-sectional, observational design and investigated the prevalence of symptoms of depression among outpatients with ARI and described the characteristics of outpatients with ARI in Kaduna State.

### Results

The prevalence of symptoms of depression was 19.6% for respondents with symptoms of ARI and 14.4% for those without symptoms of ARI. On no risk of depression, we had a higher proportion of the respondents without symptoms of ARI (86%) than those with symptoms of depression (80%) (M = 318.4, SD = 29.62 case, and M = 344.0, SD = 14.2 control, r = 0.88, CI = 13.5 to 6.5, P = 0.000952). Likewise, in the category with mild risk of depression, respondents without symptoms of ARI were fewer (10%) than those with symptoms of depression (15%) (M = 58.4, SD = 26.0 case, and M = 42.1, SD = 12.7 control, r = 0.86, CI = 11.8 to 5.8, P = 0.0136. There was no significant difference between respondents with symptoms of ARI and without symptoms of ARI in the categories of moderate (M = 13.6, SD = 5.1 case, and M = 11.6, SD = 4.6 control, r = 0.87, CI = 2.3 to 2.1, P = 0.178) and high (M =

**Funding:** The author(s) received no specific funding for this work.

**Competing interests:** The authors have declared that no competing interests exist.

**Abbreviations:** ARI, Acute Respiratory Infections; BDI, Beck's Depression Inventory; CEPI, Epidemic Preparedness and Innovation; CHEW, Community Health Environmental Worker; CHO, Community Health Officer; COVAX, Coalition for COVID-19 vaccine led by WHO; DALY, Disability Adjusted Life Year; DSM-IV, Diagnostic and Statistical Manual IV for Mental Health; LGA, Local Government Area; LMIC, Low and Middle-Income Countries; LRTI, Lower Respiratory Tract Infection; NLR, Neutrophil to Lymphocyte Ratio; NPI, Non- Pharmaceutical Interventions; PCR, Polymerase Chain Reaction; PHC, Primary Health Care; PHEIC, Public Health Emergency of International Concern; PPE, Personal Protective Equipment; PPS, Proportional Probability Size; RDT, Rapid Diagnostic Test; RSV, Respiratory Syncytial Virus; RTI, Respiratory Tract Infection; RT-PCR, Reverse Transcriptase Polymerase Chain Reaction; SARS-CoV-2, Severe Acute Respiratory Syndrome Coronavirus 2; SPSS, Statistical Package for Social Sciences; UNICEF, United Nations Children Funds; URTI, Upper Respiratory Tract Infection; WHO, World Health Organization.

5.6, SD = 2.5 case, and M = 4.4, SD = 3.2 control, r = 0.61, CI = 1.2 to 1.5, P = 0.174) risk of depression.

## Conclusion

Symptoms of depression were commoner among respondents who presented with symptoms of Acute Respiratory Infection (ARI) at the Outpatient Department (OPD). However, further explanatory research is needed to establish causality.

## Introduction

The novel SARS-CoV-2 virus that causes Coronavirus disease (COVID-19) has redefined global health and response to Acute Respiratory Infection (ARI) [1–4]. The outbreak of a cluster of influenza-like illnesses in Wuhan, China, has morphed into a pandemic in the last quarter of 2019, stretching from South East Asia to Europe, The Americas, Africa, and the Australian subcontinent [5]. The World Health Organization (WHO) declared COVID-19 a Public Health Emergency of International Concern (PHEIC) in January 2020 and a pandemic in March 2020 [6, 7]. The statistics are grim, with over 666 million laboratory-confirmed cases and 6 million deaths recorded globally as of 15th January 2023 [8]. Nigeria's first confirmed COVID-19 case was on 14th February 2020, and has recorded 266, 463 confirmed cases and 3, 155 deaths as of 15th January 2023 [9, 10]. Kaduna State, a subnational entity in federal Nigeria, had the first confirmed COVID-19 case on 28th March 2020, and a total of 11, 630 confirmed cases with 90 deaths as of 15th January 2023 [9]. The four metropolitan Local Government Areas (LGAs) of Chikun, Igabi, Kaduna North, and Kaduna South account for over 50% of all confirmed cases in Kaduna State [9].

Acute Respiratory Infection, like COVID-19, usually presents as cough, sneezing, sore throat, and difficulty breathing, with or without fever, and is transmitted through contact with an infected person or their secretions [11–13]. Other symptoms are body weakness, loss of smell, diarrhea, abdominal pains, and some non-specific symptoms [14]. The prognosis of COVID-19 is worse in the elderly and those with other co-existing conditions, for example, diabetes, hypertension, asthma, and so forth [15]. Many COVID-19 cases are asymptomatic, and laboratory test using Reverse Transcriptase Polymerase Chain Reaction (RT-PCR) and Rapid Diagnostic Test (RDT) is essential in diagnosis and case confirmation [16]. Like many low- and middle-income countries (LMIC), Nigeria has low testing rates, making COVID-19 diagnosis difficult and the efforts to contain the pandemic challenging [17].

COVID-19 now has a vaccine, but before vaccine discovery and licensing, the mainstay of response was non-pharmaceutical interventions (NPI), regular handwashing with soap under running water; respiratory hygiene, and coughing into a bent elbow; physical distancing of at least 2 meters; use of face masks; hand sanitizers; environmental sanitation; lockdown and restriction of movements and other administrative measures [18–21]. COVID-19 rollout has not been even, as the high-income countries account for over 80% of current vaccine uptake [22]. The COVAX facility, led by the World Health Organization (WHO), Coalition for Epidemic Preparedness Innovation (CEPI), and UNICEF, have supported LMICs, and the COVID-19 vaccine rollout has commenced in Nigeria and other developing countries [23].

The symptoms of depression are common, and a study by Gureje et al. has shown a prevalence of 5% among the general population in major Nigerian cities [24]. Case managers at the isolation centres have observed rising symptoms of depression among COVID-19 patients, and Sensoy et al. (2021) documented a significantly higher average level of depression (24%)

among COVID-19 patients [25]. The cardinal symptoms of depression are low mood, loss of interest in pleasurable activities, and worthlessness. Other symptoms are lack of energy, loss of appetite, body weakness, sadness, suicidal ideation, etc. COVID-19 sometimes manifests with symptoms of depression, which the isolation may worsen, restriction of movements, and lockdown in response to the COVID-19 pandemic [26].

Little or no studies have investigated the prevalence of symptoms of depression among outpatients with ARI nor described the characteristics of outpatients with ARI in Kaduna State.

The response to COVID-19 is a combination of measures ranging from enforcement of NPI, vaccination, and symptomatic and supporting treatment of patients, working with governments, NGOs, development partners, and citizens [27]. Vaccine hesitancy, low-risk perception of COVID-19 by citizens, mistrust of government, conspiracy theories, myths, and fake news have weakened compliance and enforcement of COVID-19 preventive measures.

The study described the characteristics of outpatients who presented with ARI to clinics and hospitals in four metropolitan LGAs of Kaduna State, determined the prevalence of ARI among these patients, and evaluated the relationship between ARI and symptoms of depression among the outpatients and a control group.

## Materials and methods

### Study design

We utilized a cross-sectional design, investigated the prevalence of symptoms of depression among outpatients with ARI, and described the characteristics of outpatients with ARI in Kaduna State.

### Study setting

Four (4) metropolitan Local Government Areas of Chikun, Igabi, Kaduna North, and Kaduna South were purposefully sampled because they have contributed to >50% of the total COVID-19 burden in Kaduna State since the outset of the pandemic in March 2020. Kaduna State is in Nigeria's Northwestern region, with a projected population of 9.7 million (2006 census) [28]. It was the former capital of Northern Nigeria before Independence in 1960 and home to many tribes in Nigeria. Kaduna State has 23 LGAs and 255 wards (the lowest administrative unit). The Infant Mortality rate (170/1000) and the maternal mortality rate (230/100000) are below the national average. The state health insurance scheme has enrolled about 5% of the total population (state employees), and the rest access health services through out-of-pocket expenses. The predominant tribes are Hausa and Fulani ethnic groups. Other ethnic groups are Gwari, Kadara, Bajju, Kataf, etc. Christianity, Islam, and traditional faith are the dominant religions. The state has a boundary with Kano State (north), Plateau and Nassarawa States (south), Bauchi State (east), and NigerState e(west). Kaduna State is an agricultural state and one of Nigeria's leading ginger, maize, and millet producers [28].

### Study population

The study population consisted of outpatients who presented with symptoms of ARI to primary, secondary, or tertiary health facilities, both private and public, in the four metropolitan LGAs in the last two weeks before the study and a control group.

### Sampling technique

The researcher conducted a stratified random sampling of health facilities (primary, secondary & tertiary), both private and public. The first stratum is the LGA, then the ward (lowest

administrative unit), and the type of facility (private or public). The total health facilities in the state range from Primary Health Centre (PHC) (1725), secondary facilities (29), tertiary facilities (6), and private facilities (534). The four metropolitan LGAs accounted for 13% (221) of total PHC, 20% (6) of secondary health facilities, 50% (1) of tertiary health facilities, and 33% (178) of private health facilities. A total of 33 (15%) PHCs, five (83%) secondary health facilities, one (50%) tertiary health facility, and 33 (19%) private health facilities were sampled from the four metropolitan LGAs. The sample distribution was based on probability proportional to size (PPS) (size of respondents depended on the number of outpatients with symptoms of ARI seen by the facility).

## Sample size estimation [29]

$$\text{Sample size} = \frac{Z2*(p)*(1-p)}{c2}$$

Where:
Z = Z value (e.g., 1.96 for 95% confidence level)
p = percentage picking a choice, expressed as decimal
(.5 used for sample size needed)
c = confidence interval, expressed as decimal (0.05)
= 396 (case) and 402 (control) recruited
Total participants = 798

## Study tools

**Symptoms of depression.** The researcher and his assistants utilized the Beck Depression Inventory (BDI) to measure the symptoms of depression among outpatients who presented with ARI. The BDI has undergone two major revisions: in 1978 as the BDI-IA and in 1996 as the Beck Depression Inventory-II (BDI-II). The updated BDI-II taps psychological and somatic manifestations of 2-week major depressive episodes, as operationalized in the DSM-IV.9 This version was modified to reword and replace some items. Four items of the BDI-IA that proved less sensitive for identification of typical symptoms of severe depression–weight loss, distorted body image, somatic preoccupation, and inability to work–were dropped and replaced by agitation, worthlessness, difficulty concentrating, and energy loss to assess a distinctive degree of intensity of depression. In addition, the items on appetite and sleep change were amended to evaluate the increase and decrease of these depression-related behaviors. Unlike the original version, the BDI-II does not reflect any particular theory of depression [30].

The BDI-II is a relevant psychometric instrument. It has high reliability and capacity to differentiate between depressed and non-depressed subjects and improved concurrent, content, and structural validity. And from available psychometric data, the BDI-II is a cost-effective tool for measuring the severity of depression, with good applicability for research and clinical practice worldwide. The instrument has an internal consistency of 0.9 and retest reliability ranging from 0.73 to 0.96 [30].

**ARI.** the presence of at least fever and cough and any other symptoms (sneezing, sore throat, headache, body weakness, malaise, dizziness). The ARI questionnaire also explored the duration of symptoms, type of health facility visited by the patient, clinician seen, diagnosis, recovery, and satisfaction with treatment at the health facility. The researcher and his assistants actively searched the health facility outpatients' registers for ARI [11].

**Questionnaire.** The semi-structured questionnaire has three parts: general information or biodata, a questionnaire on symptoms of depression (BDI), and a questionnaire on ARI. The questionnaire was translated to Hausa (local language) and back-translated to English to ensure internal consistency. The translator is a native Hausa speaker with over 15 years of teaching experience in Hausa and a National Certificate in Education (NCE). The interviewers received intensive 1-day training on the study rationale, questions, design, and tools.

**Procedure.** The researcher and his assistants administered the questionnaire to the randomly sampled outpatients from the outpatients' register maintained at the health facility. The interviews were conducted at the participants' homes and, in some cases, by telephone in keeping with COVID-19 protocol. The respondents were recruited and interviewed between 14th March and 15th April 2022.

We compared the study population (outpatients) with the matched control population (size, culture, locality, and socio-economic factors).

**Inclusion criteria.** All outpatients, 13 years (BDI validated in 13 years and above) and above residents in the four metropolitan LGAs who presented to the health facility in the last two weeks with ARI were sampled for the study.

**Exclusion criteria.** Outpatients with known depression and patients with debilitating illnesses.

**Data collation and statistical analysis.** The researchers used the SPSS (Statistical Package for Social Science) in data collation. Descriptive statistics for proportions and measures of central tendencies were calculated. The researcher ran a paired samples t-test and compared the means of the study and control populations at $P<0.05$. The Pearson's Coefficient of correlations was computed for the measured variable between the two groups.

## Ethical consideration

The researchers sought and obtained ethical approval (NHREC/17/03/2018) from the Kaduna State Ministry of Health.

## Written informed consent

Only respondents who willingly gave written informed consent participated in the study after a clear protocol explanation. Parents or caregivers provided consent on behalf of minors (<18 years), and assent or cooperation was obtained from the minors before the interview.

## Results

### General characteristics of the respondents

A total of 805 volunteers were recruited for the study, of which 798 (99.1%) consented to the interview, 396 (49.6%) respondents with ARI symptoms (case), and 402 (50.4%) those without ARI symptoms (control). Four (4) metropolitan LGAs of the 23 LGAs in Kaduna State (17%) participated in the study. Kaduna South has the highest number of respondents (case and control), 319 (40%), followed by Kaduna North 288 (36%), Chikun 144 (18%), and Igabi 47 (6%). A total of 33 wards (lowest administrative unit) participated in the study; Kaduna South had 13 (39%) wards, Kaduna North 12 (36%), Chikun 6 (18%), and Igabi 2 (6%). The research assistants interviewed respondents from 308 settlements out of the over 2000 settlements in the four metropolitan LGAs, with Kaduna South having the highest 107 (35%), followed by Kaduna North 99 (325), Chikun 78 (25%), and Igabi 24 (8%). The mean age of the respondents who presented with symptoms of ARI (case) was 34 years, standard deviation (SD) of 15 and a range of 13 to 90 years, and that of the

respondents without symptoms of ARI, a mean of 30, SD 12, and range 13 to 90 years. The sex distribution of the respondents was female 266 (67%), male 130 (33%), those with symptoms of ARI and female 242 (60%), male 160 (40%) those without symptoms of ARI. About 50% of the respondents have had 6 to 12 years of formal education (secondary education), case (47%), and control (53%) (see Table 1). A higher proportion of the respondents, both case and control, were unemployed (31%), followed by house-wives (27%), petty traders (26%), and the least was big-time traders (2%) (see Table 1). About 70% of the respondents, both case and control earn less than 1 USD per day, 25% make 1 USD per day, and 5% above 1 USD per day (Table 1).

**Table 1. Demographic characteristics of the respondents.**

| Characteristics | ARI Symptoms N (%) | No ARI Symptoms N (%) | [1]P-value |
|---|---|---|---|
| Age (Years) | | | P = 0.965 |
| <15 | 10 (3%) | 10 (2%) | |
| 15–19 | 47 (12%) | 51 (13%) | |
| 20–24 | 52 (13%) | 85 (21%) | |
| 25–29 | 82 (21%) | 79 (20%) | |
| 30–34 | 46 (12%) | 54 (13%) | |
| 35–39 | 38 (10%) | 48 (12%) | |
| 40–44 | 37 (9%) | 32 (8%) | |
| 45–49 | 22 (6%) | 18 (4%) | |
| 50–54 | 23 (6%) | 11 (3%) | |
| 55–59 | 13 (3%) | 4 (1%) | |
| 60–64 | 13 (3%) | 4 (1%) | |
| 65–69 | 5 (1%) | 4 (1%) | |
| >70 | 8 (2%) | 2 (0%) | |
| Level of Education[2] | | | P = 0.976 |
| Quaranic | 58 (15%) | 51 (13%) | |
| Primary | 55 (14%) | 51 (13%) | |
| Secondary | 185 (47%) | 213 (53%) | |
| Tertiary | 98 (25%) | 87 (22%) | |
| Occupation | | | P = 0.974 |
| Petty Trader | 93 (23%) | 111 (28%) | |
| Big Time Trader | 5 (1%) | 11 (3%) | |
| Private Sector Worker | 29 (7%) | 29 (7%) | |
| Government Worker | 31 (8%) | 20 (5%) | |
| House-wife | 118 (30%) | 100 (25%) | |
| Unemployed | 120 (30%) | 131 (33%) | |
| Income (USD/Day)[3] | | | P = 0.986 |
| <0.86 | 284 (72%) | 272 (68%) | |
| 0.86–1.09 | 43 (11%) | 66 (16%) | |
| 1.10–1.49 | 19 (5%) | 21 (5%) | |
| 1.50–1.69 | 17 (4%) | 11 (3%) | |
| >1.70 | 33 (8%) | 32 (8%) | |

[1]Statistical significance at p<0.05 (2-tailed)

[2]Quaranic = 2 years schooling; Primary = 6years; Secondary = 9-12years; Tertiary = 16years and above

[3]Prevailing exchange rate of 580 Naira to 1 USD

## Features of respondents presenting with symptoms of acute respiratory infections

The most frequent symptom among those with ARI was sore throat (25%); others were fever (21%), cough (20%), headache (14%), and the minor symptoms are abdominal pain, catarrh, dizziness, heartburn, and nausea (<1%) (Fig 1). The average duration of the symptoms was 3–4 days (46%), symptoms <14 days (93%), and >14 days (7%). The majority of the respondents with symptoms of ARI presented to the Primary Healthcare Centre (PHC) (79%) for treatment. Corollary, many of the respondents who visited PHC were attended to by Community Health Extension Workers (CHEW) (50%), Community Health Officers (CHO) (18%), and at other facilities by medical officers (19%), and least consultant physician (4%). The most typical diagnoses among respondents who presented with symptoms of ARI were malaria (45%) and Typhoid fever (28%), and the least were COVID-19 and diabetes mellitus (<1%) (Fig 2). Most of the respondents who presented with symptoms of ARI were satisfied (97%) with the treatment received at the health facilities visited, and 89% of the respondents who sought treatment recovered after completing their treatment. The average recovery days were 3–4 days (44%), and 10% of the respondents stayed seven days and above before recovery from ARI symptoms. Among the respondents, 31% had a history of a family member presenting

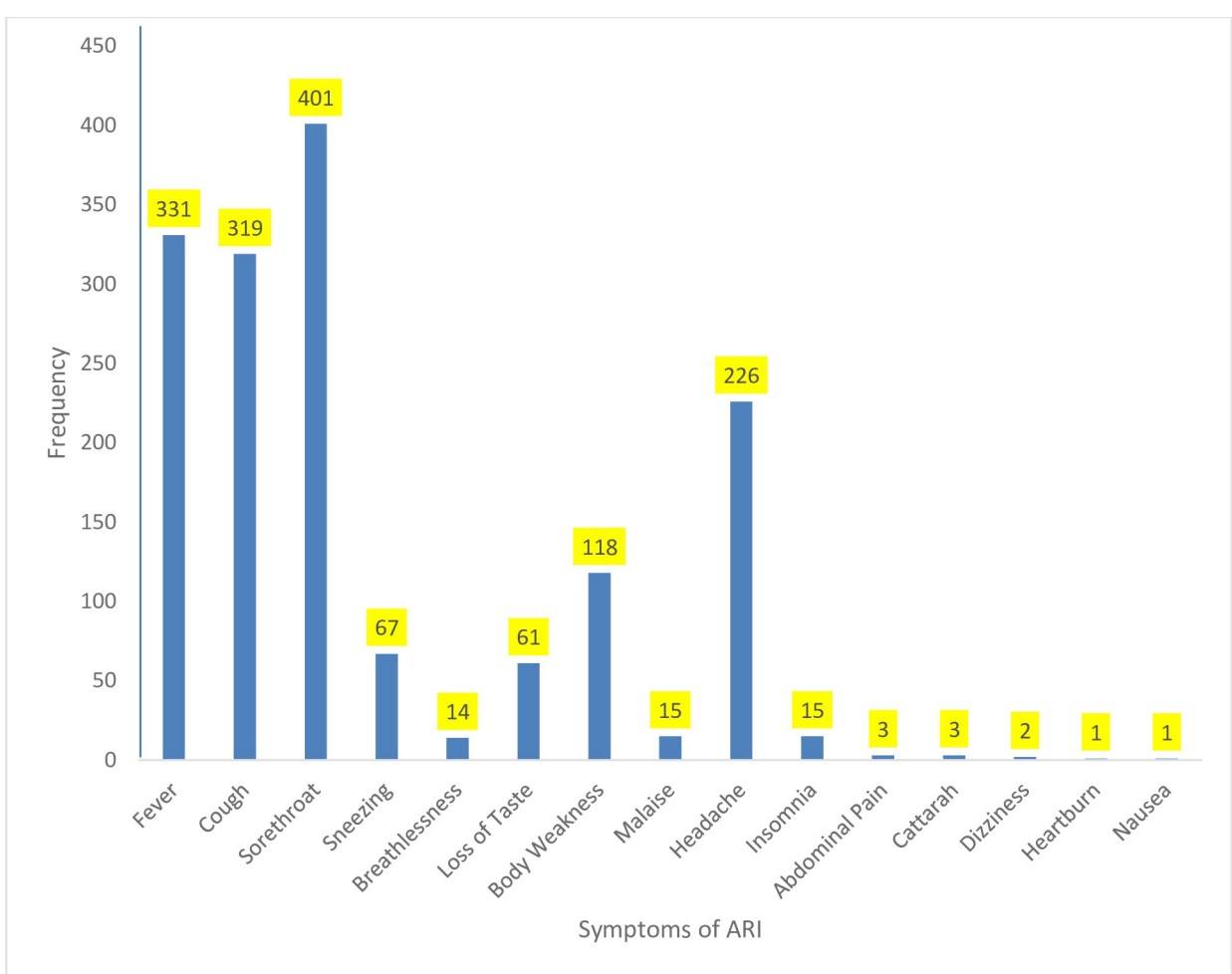

**Fig 1.** Symptoms of ARI among respondents who presented at the outpatient department of health facility with symptoms of ARI.

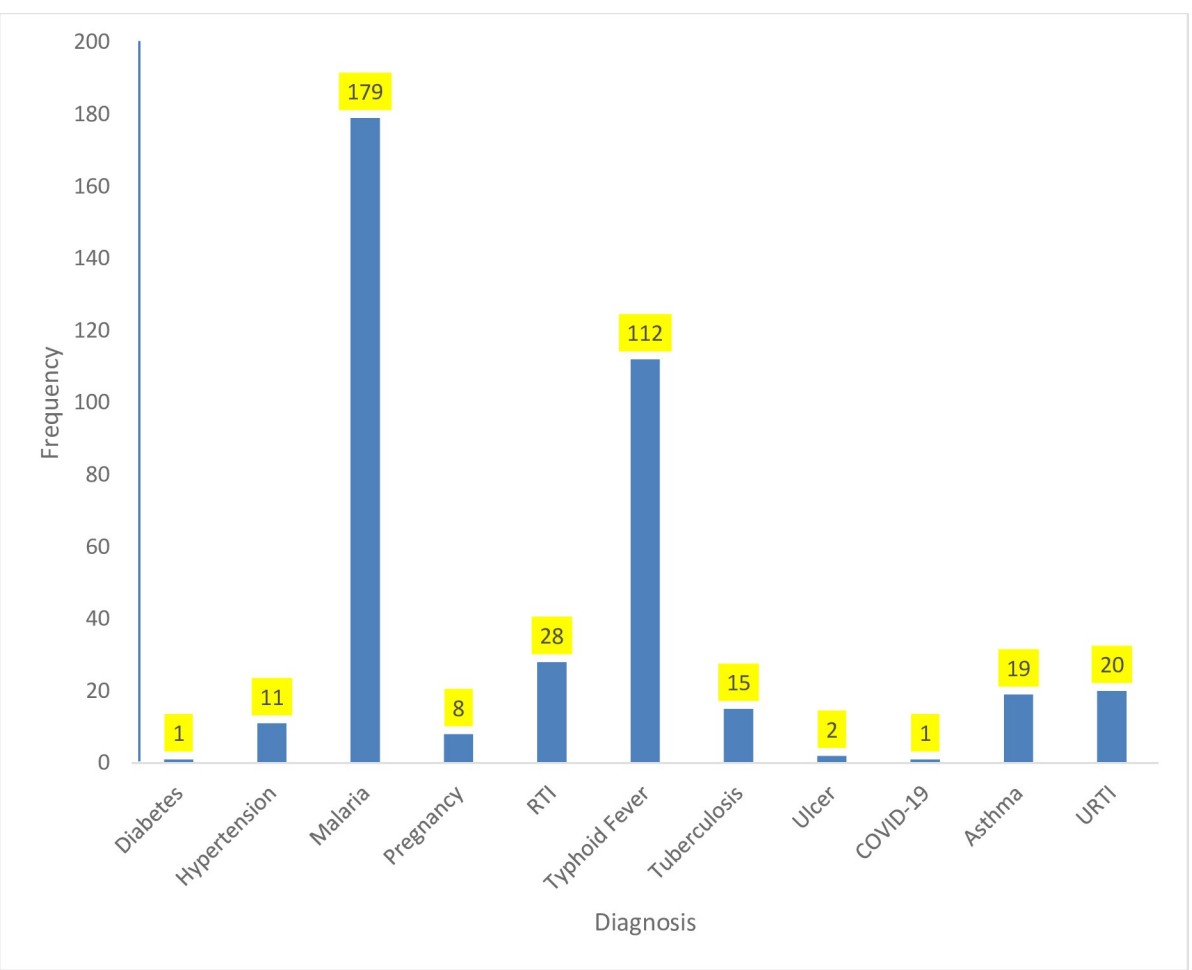

**Fig 2. Diagnosis by clinicians of respondents who presented at the outpatient department of health facility with symptoms of ARI.**

with ARI symptoms. Likewise, 35% of the respondents knew somebody outside the family who presented with symptoms of ARI (Table 2).

### Respondents' scores on Beck's Depression Inventory

The Beck Depression Inventory (BDI) classified the respondents into four categories (no risk of depression, mild, moderate, and high risk of depression) based on respondents' scores on the items (Table 3). On no risk of depression, we had a higher proportion of the respondents without symptoms of ARI (86%) than those with symptoms of depression (80%) (M = 318.4, SD = 29.62 case, and M = 344.0, SD = 14.2 control, r = 0.88, CI = 13.5 to 6.5, P = 0.000952). Likewise, in the category with mild risk of depression, respondents without symptoms of ARI were fewer (10%) than those with symptoms of depression (15%) (M = 58.4, SD = 26.0 case, and M = 42.1, SD = 12.7 control, r = 0.86, CI = 11.8 to 5.8, P = 0.0136. There was no significant difference between respondents with symptoms of ARI and without symptoms of ARI in the categories of moderate (M = 13.6, SD = 5.1 case, and M = 11.6, SD = 4.6 control, r = 0.87, CI = 2.3 to 2.1, P = 0.178) and high (M = 5.6, SD = 2.5 case, and M = 4.4, SD = 3.2 control, r = 0.61, CI = 1.2 to 1.5, P = 0.174) risk of depression. The prevalence of symptoms of depression was 19.6% (case)

**Table 2. Features of respondents presenting with symptoms of Acute Respiratory Infections (ARI).**

|  | N | % |
|---|---|---|
| Symptoms |  |  |
| Fever | 331 | 21% |
| Cough | 319 | 20% |
| Sorethroat | 401 | 25% |
| Sneezing | 67 | 4% |
| Breathlessness | 14 | 1% |
| Loss of Taste | 61 | 4% |
| Body Weakness | 118 | 7% |
| Malaise | 15 | 1% |
| Headache | 226 | 14% |
| Insomnia | 15 | 1% |
| Abdominal Pain | 3 | 0% |
| Cattarah | 3 | 0% |
| Dizziness | 2 | 0% |
| Heartburn | 1 | 0% |
| Nausea | 1 | 0% |
| Symptoms Duration |  |  |
| >14 days | 28 | 7% |
| 10–14 days | 27 | 7% |
| 1–2 days | 52 | 13% |
| 3-4days | 183 | 46% |
| 5–6 days | 79 | 20% |
| 7–10 days | 27 | 7% |
| Facility Visited |  |  |
| Primary (PHC) | 312 | 79% |
| Private Hospital | 45 | 11% |
| Secondary (General Hospital) | 32 | 8% |
| Tertiary (Teaching Hospital) | 7 | 2% |
| Clinician Seen |  |  |
| CHEW[1] | 197 | 50% |
| CHO[2] | 73 | 18% |
| Consultant Physician | 16 | 4% |
| Medical Officer | 77 | 19% |
| RN[3] | 33 | 8% |
| Clinical Diagnosis |  |  |
| Diabetes | 1 | 0% |
| Hypertension | 11 | 3% |
| Malaria | 17 | 45% |
| Pregnancy | 8 | 2% |
| LRTI[4] | 28 | 7% |
| Typhoid Fever | 112 | 28% |
| Tuberculosis | 15 | 4% |
| Ulcer | 2 | 1% |
| COVID-19 | 1 | 0% |
| Asthma | 19 | 5% |
| URTI[5] | 20 | 5% |
| Satisfied with Treatment |  |  |

*(Continued)*

**Table 2.** (Continued)

|  | N | % |
|---|---|---|
| No | 11 | 3% |
| Yes | 385 | 97% |
| Recovered after Treatment |  |  |
| No | 44 | 11% |
| Yes | 352 | 89% |
| Recovery Days |  |  |
| 10–14 days | 13 | 4% |
| 1–2 days | 110 | 31% |
| 3-4days | 155 | 44% |
| 5–6 days | 54 | 15% |
| 7–10 days | 20 | 6% |
| Family History of Similar Illness |  |  |
| No | 272 | 69% |
| Yes | 124 | 31% |
| Know Anybody with Similar Illness |  |  |
| No | 258 | 65% |
| Yes | 138 | 35% |

[1]Community Health Environmental Worker

[2]Community Health Officer

[3]Ragistered Nurse

[4]Lower Respiratory Tract Infection

[5]Upper Respiratory Tract Infection

and 14.4% (control), and by risk classification, mild (case = 15%, control = 10%), moderate (case = 3%, control = 3%), and high (case = 1%, control = 1%) (Table 4).

**Association between symptoms of ARI and depression.** The commonest symptoms of depression among the respondents in both case and control were sleep disturbances, feelings of sadness, easy tiredness, and low appetite, of which the majority were female; I don't sleep well (male 38%, female 62%), I feel sad (male 37%, female 63%), easy tiredness (male 33%, female 67%), and low appetite (male 33%, female 67%).

The respondents who were unemployed (33%) and house-wives (30%) accounted for the commonest symptoms of depression (feeling of sadness, easy tiredness, disturbed sleep, and low appetite) observed among all respondents, both those with symptoms of ARI and those without the symptoms of ARI, while government workers (7%) and big-time traders (2%) had the least symptoms of depression.

The respondents who had diabetes and typhoid fever accounted for the most typical symptoms of depression (feeling of sadness, easy tiredness, disturbed sleep, and low appetite) observed among all respondents, both those with symptoms and without symptoms of ARI and those with hypertension, URTI, and RTI had the least symptoms of depression.

## Discussion

This study has shown that the likelihood of no symptoms or mild symptoms of depression is commoner among respondents with no symptoms of Acute Respiratory Infection (ARI). Though causality cannot be inferred, the association is statistically significant (see Table 4). However, there was no significant difference between respondents with ARI symptoms and no

**Table 3. Distribution of respondents' scores on Beck's Depression Inventory.**

| Items | ARI Symptoms N (%) | No ARI Symptoms N (%) |
|---|---|---|
| 0. I am no more irritated by things than I ever was. | 327 (49%) | 342 (51%) |
| 0. I am no more worried about my health than usual. | 310 (48%) | 341 (52%) |
| 0. I can sleep as well as usual. | 272 (46%) | 323 (54%) |
| 0. I can work about as well as before. | 300 (47%) | 341 (53%) |
| 0. I don't cry any more than usual. | 338 (49%) | 347 (51%) |
| 0. I don't feel I am any worse than anybody else. | 341 (49%) | 358 (51%) |
| 0. I don't feel I am being punished. | 343 (49%) | 356 (51%) |
| 0. I don't feel that I look any worse than I used to. | 345 (49%) | 358 (51%) |
| 0. I don't get more tired than usual. | 259 (45%) | 323 (55%) |
| 0. I don't have any thoughts of killing myself. | 357 (49%) | 365 (51%) |
| 0. I have not noticed any recent change in my interest in sex. | 290 (47%) | 323 (53%) |
| 0. I make decisions about as well as I ever could. | 344 (49%) | 352 (51%) |
| 0. I am not particularly discouraged about the future. | 336 (49%) | 350 (51%) |
| 0. I do not feel like a failure. | 344 (49%) | 353 (51%) |
| 0. I do not feel sad. | 283 (47%) | 315 (53%) |
| 0. I don't feel disappointed in myself. | 352 (49%) | 365 (51%) |
| 0. I don't feel particularly guilty | 330 (48%) | 352 (52%) |
| 0. I get as much satisfaction out of things as I used to. | 308 (47%) | 341 (53%) |
| 0. I have not lost interest in other people. | 332 (50%) | 336 (50%) |
| 0. I haven't lost much weight, if any, lately. | 304 (47%) | 347 (53%) |
| 0. My appetite is no worse than usual. | 271 (45%) | 335 (55%) |
| 1. I am critical of myself for my weaknesses or mistakes. | 41 (58%) | 30 (42%) |
| 1. I am disappointed in myself. | 33 (56%) | 26 (44%) |
| 1. I am less interested in other people than I used to be. | 48 (49%) | 50 (51%) |
| 1. I am less interested in sex than I used to be. | 71 (60%) | 48 (40%) |
| 1. I am slightly more irritated now than usual. | 54 (54%) | 46 (46%) |
| 1. I am worried about physical problems like aches, pains, upset stomach, or constipation. | 57 (60%) | 38 (40%) |
| 1. I am worried that I am looking old or unattractive. | 36 (55%) | 29 (45%) |
| 1. I cry more now than I used to. | 34 (50%) | 34 (50%) |
| 1. I don't enjoy things the way I used to. | 73 (61%) | 47 (39%) |
| 1. I don't sleep as well as I used to. | 90 (61%) | 57 (39%) |
| 1. I feel I may be punished. | 34 (54%) | 29 (46%) |
| 1. I get tired more easily than I used to. | 112 (64%) | 62 (36%) |
| 1. I have lost more than five pounds. | 74 (65%) | 40 (35%) |
| 1. I have thoughts of killing myself, but I would not carry them out. | 27 (50%) | 27 (50%) |
| 1. I put off making decisions more than I used to. | 34 (49%) | 36 (51%) |
| 1. It takes an extra effort to get started at doing something. | 72 (64%) | 41 (36%) |
| 1. My appetite is not as good as it used to be. | 107 (66%) | 56 (34%) |
| 1. I feel discouraged about the future. | 44 (51%) | 42 (49%) |
| 1. I feel guilty a good part of the time. | 51 (59%) | 35 (41%) |
| 1. I feel I have failed more than the average person. | 38 (51%) | 36 (49%) |
| 1. I feel sad | 96 (56%) | 75 (44%) |
| 2. I am disgusted with myself. | 8 (53%) | 7 (47%) |
| 2. I am quite annoyed or irritated a good deal of the time. | 11 (52%) | 10 (48%) |
| 2. I am very worried about physical problems, and it's hard to think of much else. | 23 (55%) | 19 (45%) |

*(Continued)*

**Table 3.** (Continued)

| Items | ARI Symptoms N (%) | No ARI Symptoms N (%) |
|---|---|---|
| 2. I blame myself all the time for my faults. | 11 (52%) | 10 (48%) |
| 2. I cry all the time now. | 11 (65%) | 6 (35%) |
| 2. I don't get real satisfaction out of anything anymore. | 11 (52%) | 10 (48%) |
| 2. I expect to be punished. | 14 (54%) | 12 (46%) |
| 2. I feel there are permanent changes in my appearance that make me look unattractive | 11 (55%) | 9 (45%) |
| 2. I get tired from doing almost anything. | 17 (52%) | 16 (48%) |
| 2. I have almost no interest in sex. | 26 (53%) | 23 (47%) |
| 2. I have greater difficulty in making decisions more than I used to. | 13 (50%) | 13 (50%) |
| 2. I have lost more than ten pounds. | 13 (52%) | 12 (48%) |
| 2. I have lost most of my interest in other people. | 12 (52%) | 11 (48%) |
| 2. I have to push myself very hard to do anything. | 18 (50%) | 18 (50%) |
| 2. I wake up 1–2 hours earlier than usual and find it hard to get back to sleep. | 24 (63%) | 14 (37%) |
| 2. I would like to kill myself. | 8 (73%) | 3 (27%) |
| 2. My appetite is much worse now. | 13 (59%) | 9 (41%) |
| 2. As I look back on my life, all I can see is a lot of failures. | 11 (50%) | 11 (50%) |
| 2. I am sad all the time and I can't snap out of it. | 12 (57%) | 9 (43%) |
| 2. I feel I have nothing to look forward to. | 9 (50%) | 9 (50%) |
| 2. I feel quite guilty most of the time. | 10 (45%) | 12 (55%) |
| 3. I am dissatisfied or bored with everything. | 4 (50%) | 4 (50%) |
| 3. I am so worried about my physical problems that I cannot think of anything else. | 6 (60%) | 4 (40%) |
| 3. I am too tired to do anything. | 8 (89%) | 1 (11%) |
| 3. I believe that I look ugly. | 4 (40%) | 6 (60%) |
| 3. I blame myself for everything bad that happens. | 3 (43%) | 4 (57%) |
| 3. I can't do any work at all. | 6 (75%) | 2 (25%) |
| 3. I can't make decisions at all anymore. | 5 (83%) | 1 (17%) |
| 3. I feel I am being punished. | 5 (50%) | 5 (50%) |
| 3. I feel irritated all the time. | 4 (50%) | 4 (50%) |
| 3. I hate myself. | 3 (43%) | 4 (57%) |
| 3. I have lost all of my interest in other people. | 4 (44%) | 5 (56%) |
| 3. I have lost interest in sex completely. | 9 (53%) | 8 (47%) |
| 3. I have lost more than fifteen pounds. | 5 (63%) | 3 (38%) |
| 3. I have no appetite at all anymore. | 5 (71%) | 2 (29%) |
| 3. I used to be able to cry, but now I can't cry even though I want to. | 13 (46%) | 15 (54%) |
| 3. I wake up several hours earlier than I used to and cannot get back to sleep. | 10 (56%) | 8 (44%) |
| 3. I would kill myself if I had the chance. | 4 (36%) | 7 (64%) |
| 3. I am so sad and unhappy that I can't stand it. | 5 (63%) | 3 (38%) |
| 3. I feel guilty all of the time. | 5 (63%) | 3 (38%) |
| 3. I feel I am a complete failure as a person. | 3 (60%) | 2 (40%) |
| 3. I feel the future is hopeless and that things cannot improve. | 7 (88%) | 1 (13%) |

ARI symptoms on moderate to high risk of the symptoms of depression. The respondents with symptoms of ARI had a higher prevalence of symptoms of depression (19.6%), compared to those with no symptoms of ARI (14.4%).

**Table 4. A comparison of mean, standard deviation, and P-value scores of respondents on the Beck Depression Inventory and the prevalence of symptoms of depression.**

| | Mean | SD | P-value | r |
|---|---|---|---|---|
| No Risk of Depression[1] | | | | |
| ARI Symptoms (Case) | 318.4 | 29.6 | 0.000952 | 0.882* |
| No ARI Symptoms (Control) | 344.0 | 14.2 | | |
| Mild risk of Depression[2] | | | | |
| ARI Symptoms (Case) | 58.4 | 26.0 | 0.0135 | 0.863* |
| No ARI Symptoms (Control) | 42.1 | 12.7 | | |
| Moderate Risk of Depression[3] | | | | |
| ARI Symptoms (Case) | 13.6 | 5.1 | 0.178 | 0.866* |
| No ARI Symptoms (Control) | 11.6 | 4.6 | | |
| High Risk of Depression[4] | | | | |
| ARI Symptoms (Case) | 5.6 | 2.5 | 0.174 | 0.612* |
| No ARI Symptoms (Control) | 4.4 | 3.2 | | |
| Prevalence of Symptoms of Depression | | | | |
| ARI Symptoms (Case) | 19.6% (1630/8316) | | | 0.122* |
| No ARI Symptoms (Control) | 14.4% (1219/8442) | | | |
| Prevalence by Risk Classification | | | | |
| Mild | | | | |
| Case | 15% (1226/8316) | | | |
| Control | 10% (884/8442) | | | |
| Moderate | | | | |
| Case | 3% (286/8316) | | | |
| Control | 3% (243/8442) | | | |
| High | | | | |
| Case | 1% (118/8316) | | | |
| Control | 1% (92/8442) | | | |

* Pearson's coefficient of correlation at P<0.05

[1]Cumulative scores on all items (questionnaires) starting with 0

[2]Cumulative scores on all items (questionnaires) starting with 1

[3]Cumulative scores on all items (questionnaires) starting with 2

[4]Cumulaive scores on all items (questionnaires) starting with 3

The symptoms of ARI were prevalent among the respondents who presented as outpatients at private and public health facilities. The most typical diagnoses were malaria and typhoid fever. The most familiar symptoms of ARI are sore throat, fever, and cough. The diagnosis of COVID-19, one of the differential diagnoses of ARI based purely on symptomatology, can be misleading because many common conditions (malaria, typhoid fever) present with similar symptoms.

This study has also shown that the unemployed and house-wives and those with diagnoses of diabetes and typhoid fever accounted for a higher proportion of the symptoms of depression, such as the feeling of sadness, easy tiredness, sleep disturbance, low appetite, and commoner among female than male in both respondents with ARI symptoms and without ARI symptoms. However, this pattern may reflect the higher percentage of females (64%) among the total case and control respondents. These findings are consistent with previous studies that demonstrated a high prevalence of the symptoms of depression and features of other mental illnesses among clients who presented with symptoms of ARI [24].

## Implications

Since the symptoms of depression can manifest in common conditions like malaria, typhoid fever, diabetes, and hypertension, clinicians should widen their scope when profiling clients for the risk of depression. The diagnosis of COVID-19 requires detailed history taking, with particular attention to onset, duration, and severity of symptoms, travel history, and contact with people suffering from symptoms suggestive of COVID-19 or diagnosis of COVID-19.

## Local and regional implications

The decline in the incidence of COVID-19 should be sustained through reinforced risk communication, community engagement, and enforcement of non-pharmaceutical interventions, like social distancing, handwashing with soap under running water, and use of facemasks, environmental sanitation, and personal hygiene. The political leadership at the local and regional levels working through the relevant response organs (Nigeria's Presidential Taskforce and African Union's Centre for Disease Control and Prevention) should engage and collaborate with traditional and religious institutions and their leadership, and other professional bodies and institutions to deepen awareness of COVID-19 infections, its transmission, and prevention measures, and clear all myths on COVID-19 vaccination. This engagement will promote the uptake of the COVID-19 vaccine, especially among the elderly and the vulnerable population (those living with co-morbidity, like diabetes and hypertension).

The local and regional leaders should also engage political leaders at all levels through periodic physical and virtual meetings and interactions to identify successes and challenges and provide policy direction, resources, and workforce to overcome the economic, social, medical, and personal losses to the COVID-19 pandemic, especially in developing countries like Nigeria.

## International implications

The World Health Organization (WHO) should continue to lead the coordinated response to COVID-19 through collaboration with national governments, development partners, and donors. WHO should sustain her leadership role, policy, technical assistance, research and advocacy, and resource mobilization in the face of the rampaging COVID-19 pandemic. The WHO, working with other development partners and donors, should support low and middle-income countries, especially in Africa and Asia, with policy and technical assistance, free COVID-19 vaccines, PPEs, and financial grants to enable fragile states to cope with the unprecedented economic, social, and personal losses to COVID-19 pandemic.

The one health approach through sustained communication, collaboration, and coordination of human health, animal health, and environmental interventions remains the proven response to the COVID-19 pandemic and other emerging and re-emerging diseases.

## Recommendations

The study has shown more respondents with symptoms of ARI visited the Primary Healthcare Centers (PHC), where most health workers have limited skills and training in treating mental illnesses, such as depression. Having psychiatrists, psychologists, and mental health specialists posted to PHCs in developing countries may be challenging; however, the Community Health Officers (CHO) and Community Health Extension Workers (CHEW) can receive sufficient training in mental health to enable them to profile all OPD patients for depression and other mental illnesses and refer clients with symptoms of depression to appropriate facilities for better care.

Establishing a robust, efficient, and effective referral system, with the protocol fully communicated to health workers and managers at the different levels of care, will reduce delays in the referral of clients and improve the overall care of clients with mental illnesses like depression.

The high prevalence of symptoms of depression among those with symptoms of ARI and those without symptoms of ARI demands more attention and resources to diagnose and care for clients with mental illnesses that may be co-morbid with other common conditions, like malaria, typhoid fever, diabetes, hypertension and so forth. The leadership of developing countries should allocate more resources to health and a significant proportion to address the high burden of mental illness that accounts for colossal Disability Adjusted Life Years (DALYs) in most Low- and Middle-Income Countries (LMICs), like Nigeria [31].

The education and sensitization of clients and the public will increase the awareness of symptoms of depression, improve the health-seeking behaviors of the citizens, lead to early presentation to health facilities, and reduce visits to quacks and unlicensed mental health practitioners.

## Limitations

The study was cross-sectional, and no causality was inferred. ARI is not a cause of symptoms of depression. The study was conducted in only four (4) out of the 23 LGAs of Kaduna State, though the sample size was significant, and the findings can be generalized to the total population. Some of the BDI items or questionnaires tapped on profoundly personal and relationship experiences that may provoke pain or trauma; though un-intended, the researcher and his assistants obtained informed consent from respondents after counseling and adequately explained the nature, objectives, and scope of the study in the preferred language. Recall bias was considered a limitation as participants were to provide information on the health occurrence in the two weeks before data collection.

## The need for further research

Further research is needed to fully explore the relationship between common medical conditions, for example, malaria, typhoid fever, diabetes, and symptoms of depression and other mental health illnesses. The prevalence of other mental illnesses, like anxiety disorders, should be investigated, and results should be disseminated to health workers and managers to guide clients' profiling, care, and management appropriately.

## Conclusion

Though symptoms of depression were common among respondents who presented with symptoms of Acute Respiratory Infection (ARI) at the Outpatient Department (OPD), causality cannot be inferred. Many common conditions, like malaria, typhoid fever, diabetes, and hypertension, can manifest with symptoms of depression, and clinicians can miss the diagnosis of mental illness like depression if other distinguishing features of depression are not elicited through detailed history taking, examination, and investigation by qualified and trained health workers. The burden of mental illness, like depression, is enormous and accounts for many Disability Adjusted Life Years (DALYs) in LMICs. More resources, a trained and motivated workforce, and other support to diagnose, care, and manage clients presenting with symptoms of depression and other mental illnesses in private and public health facilities are needed, especially in LMICs.

## Supporting information

**S1 Data.**
(XLSX)

**S1 Questionnaire.**
(XLSX)

## Acknowledgments

We are grateful to the research assistants (Grace Barnabas, Rebecca Boniface, Hafsat Aliyu, Khadija Ahmed, Amina Umar Madaki, Halima Danjuma, Alice Itodo, Bilikisu Baba, Maryam Ahmed, Zafira Ismail, Alheri John, Aisha Inuwa, Habiba Suleiman, Aisha Ibrahim, and Ann Utomi), who facilitated respondents' interviews and data collection, and others for their support and efforts in preparing this manuscript.

## Author Contributions

**Conceptualization:** Gregory C. Umeh, Laurent Cleenwerck de Kiev.

**Methodology:** Jabani Mamza, Aliyu Atiku, Suleiman Mohammed, Dauda S. Hananiya, Moses Onoh, Habibu B. Yahaya, Basirat Adeoti, Rabiat T. Musa, Mutiu Adegbite, Sunday Audu, Jeremiah Daikwo, Neyu Iliyasu, Amina Mohammed Baloni.

**Writing – original draft:** Gregory C. Umeh.

**Writing – review & editing:** Gregory C. Umeh, Laurent Cleenwerck de Kiev, Jabani Mamza, Aliyu Atiku, Suleiman Mohammed, Dauda S. Hananiya, Moses Onoh, Habibu B. Yahaya, Basirat Adeoti, Rabiat T. Musa, Mutiu Adegbite, Sunday Audu, Jeremiah Daikwo, Neyu Iliyasu, Amina Mohammed Baloni.

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
