## [Decision Letter · Decision Letter 0]

22 Jan 2024

PONE-D-23-18154Symptoms of Depression among Outpatients with Acute Respiratory Infection in Metropolitan Local Government Areas of Kaduna State, NigeriaPLOS ONE

Dear Dr. Umeh,

Thank you for submitting your manuscript to PLOS ONE. After careful consideration, we feel that it has merit but does not fully meet PLOS ONE’s publication criteria as it currently stands. Therefore, we invite you to submit a revised version of the manuscript that addresses the points raised during the review process. In addition to comments provided below by the 2 reviewers, please address the following comments: **Data availability: **please indicate why data is only available on request. Earlier, it was stated that data is available without restriction.**Abstract:** unclear why “future tense” is used for material and methods; this is a study with results and not a protocol.**Methods:** Sampling technique - stratified random sampling technique was used. Please define the strata and be explicit about this in relation to the PPS.**Study tools (line 167) **– the ARI screening using fever and cough and any of the other symptoms listed, was it a standard screening protocol for ARI? If yes, please provide reference.**Informed consents:** 13yrs are considered minors and consents for them should be different including interviews. Please provide details.**Results:**Lines 226 – 228: visits to primary versus tertiary healthcare facilities – were equal numbered sampled from these 2 categories of facilities and then screened for ARI? If not, then this statement should be adjusted.

**Recommendation:**

Since no causality is inferred, the question readers may have is why trying to identify depression among patients with symptom of ARI? Why not patients with malaria or diabetes or any other illness? The community health officers can be trained to screen all OPD patients for depression.

**General observations:**

It is difficult to understand the intention of the authors about whether the manuscript is about C19 or other ARI. So much about C19 in introduction and 75% of the discussion is about C19 response, locally and internationally. BTW, C19 is not even part of the title. Though C19 is ARI, but it deserves to be mentioned in the title since it is a special ARI and the authors have devoted almost 75% of the paper to C19.

Kind regards,

Ibrahim Jahun, MD, MSC, PhD

Academic Editor

PLOS ONE

2. In the online submission form, you indicated that "data available on request to the corresponding author". 

**Reviewers' comments:**

Reviewer's Responses to Questions

**Comments to the Author**

1. Is the manuscript technically sound, and do the data support the conclusions?

Reviewer #1: Yes

Reviewer #2: Yes

2. Has the statistical analysis been performed appropriately and rigorously? 

Reviewer #1: Yes

Reviewer #2: Yes

3. Have the authors made all data underlying the findings in their manuscript fully available?

Reviewer #1: Yes

Reviewer #2: Yes

4. Is the manuscript presented in an intelligible fashion and written in standard English?

Reviewer #1: Yes

Reviewer #2: Yes

5. Review Comments to the Author

Reviewer #1: The article is very useful literary material. It will be clearer if other underlying illnesses such as cardiovascular, disease, asthma and diabetes were used in classifying the subjects, beyond just cases and controls. ARI might not necessarily be the immediate cause of depression.

Reviewer #2: The manuscript has addressed an important research question considering the health indices of the research setting and the socioeconomic status of the study population. The manuscript is also well written in standard academic writing style. However, there are a few concerns the authors need to revise, which are stated below:

ABSTRACT

Background: The background should clearly state the research question, which is the prevalence of depression among patients diagnosed with ARI.

MATERIAL AND METHODS

This section should state the measuring instruments used in this study and the study period. A summary of the statistical analysis deployed should also be stated. The research question should be removed from this section and taken to the background.

RESULTS

The results is well written. However, the results may be revised to show the proportion of patients that had depression in each class. This way, we should be able to see that people with ARI were more likely to be depressed than those without ARI. It is good that the authors emphasized that causation is not implied by drawing a correlation between ARI and depression.

CONCLUSION

The conclusion is well written. The authors achieved the aim of the study.

INTRODUCTION

The introduction should be revised to show the risks and dangers of depression, which justify the study. There is a lot of talk about COVID, ARI, and the epidemiology of COVID. However, the central issue in this study is depression. Why are the authors worried about depression among patients with ARI given that only 5% of people in the general population in Kaduna have this problem?

MATERIAL and METHODS

Please describe the elements of the questionnaire and if possible, the questionnaire should be included as appendix or supplementary document.

Indicate if the translators of the questionnaire from English to Hausa were professionals. Also indicate the training the interviewer received.

6. PLOS authors have the option to publish the peer review history of their article (what does this mean?). If published, this will include your full peer review and any attached files.

Reviewer #1: **Yes: **Ado Garba Abubakar

Reviewer #2: **Yes: **Dr. Yohanna Kambai Avong

---

## [Author Response · Author response to Decision Letter 0]

31 Jan 2024

Dear Editor,

Thank you for the excellent feedback.

We have revised the manuscript thoroughly and responded to all queries from the editor and reviewers.

Umeh Gregory

---

## [Decision Letter · Decision Letter 1]

20 Feb 2024

PONE-D-23-18154R1Symptoms of Depression among Outpatients with Acute Respiratory Infection (COVID-19) in Metropolitan Local Government Areas of Kaduna State, NigeriaPLOS ONE

Dear Dr. Umeh,

Thank you for submitting your manuscript to PLOS ONE. After careful consideration, we feel that it has merit but does not fully meet PLOS ONE’s publication criteria as it currently stands. Therefore, we invite you to submit a revised version of the manuscript that addresses the points raised during the review process.

**Thank you for responding to all comments provided by the Academic Editor and the 2 reviewers. The response provided are acceptable and have addressed most of the issues raised during the review process. However, the edit provided in the title is unacceptable. The revised title reads "Symptoms of Depression among Outpatients with Acute Respiratory Infection *(COVID-19)* in Metropolitan Local Government Areas of Kaduna State, Nigeria.**

**Please note that COVID-19 is not synonymous with ARI. The patients have not been confirmed to have COVID-19, they are just presenting with symptoms that may or may not be COVID-19. However, since the period was during COVID-19 pandemic, then they apparently they were COVID-19 suspects. Therefore the tile should be something like *"Symptoms of Depression among Outpatients with suspected COVID-19 in Metropolitan Local Government Areas of Kaduna State, Nigeria). *Please note that this is just a suggestion to guide the authors and they are not mandated to use this title.**

**We look forward to receiving your revised manuscript.**

We look forward to receiving your revised manuscript.

Kind regards,

Ibrahim Jahun, MD, MSC, PhD

Academic Editor

PLOS ONE

Journal Requirements:

Reviewers' comments:

Reviewer's Responses to Questions

**Comments to the Author**

1. If the authors have adequately addressed your comments raised in a previous round of review and you feel that this manuscript is now acceptable for publication, you may indicate that here to bypass the “Comments to the Author” section, enter your conflict of interest statement in the “Confidential to Editor” section, and submit your "Accept" recommendation.

Reviewer #1: All comments have been addressed

Reviewer #2: All comments have been addressed

2. Is the manuscript technically sound, and do the data support the conclusions?

Reviewer #1: Yes

Reviewer #2: Yes

3. Has the statistical analysis been performed appropriately and rigorously? 

Reviewer #1: Yes

Reviewer #2: Yes

4. Have the authors made all data underlying the findings in their manuscript fully available?

Reviewer #1: Yes

Reviewer #2: Yes

5. Is the manuscript presented in an intelligible fashion and written in standard English?

Reviewer #1: Yes

Reviewer #2: Yes

6. Review Comments to the Author

Reviewer #1: The revised version looks good. It has addressed my questions in the previous version, particularly under the listed limitations by the authors.

Reviewer #2: (No Response)

7. PLOS authors have the option to publish the peer review history of their article (what does this mean?). If published, this will include your full peer review and any attached files.

Reviewer #1: **Yes: **Ado Garba Abubakar

Reviewer #2: **Yes: **Yohanna Kambai Avong

---

## [Author Response · Author response to Decision Letter 1]

22 Feb 2024

The authors have revised the manuscript, and are grateful to the editor and reviewers for their support.

---

## [Editor Report · Decision Letter 2]

26 Feb 2024

Symptoms of Depression among Outpatients with Suspected COVID-19 in Metropolitan Local Government Areas of Kaduna State, Nigeria

PONE-D-23-18154R2

Dear Dr. Umeh,

We’re pleased to inform you that your manuscript has been judged scientifically suitable for publication and will be formally accepted for publication once it meets all outstanding technical requirements.

Kind regards,

Ibrahim Jahun, MD, MSC, PhD

Academic Editor

PLOS ONE
---

## [Editor Report · Acceptance letter]

18 Mar 2024

PONE-D-23-18154R2 

PLOS ONE

Dear Dr. Umeh, 

I'm pleased to inform you that your manuscript has been deemed suitable for publication in PLOS ONE. Congratulations! Your manuscript is now being handed over to our production team.

Kind regards, 

on behalf of

Dr. Ibrahim Jahun 

Academic Editor

PLOS ONE